# Rational Design of Topical Semi-Solid Dosage Forms-How Far Are We?

**DOI:** 10.3390/pharmaceutics15071822

**Published:** 2023-06-26

**Authors:** Michael E. Herbig, Dirk-Heinrich Evers, Sascha Gorissen, Melanie Köllmer

**Affiliations:** RaDes GmbH, Schnackenburgallee 114, 22525 Hamburg, Germany

**Keywords:** semi-solid formulation, topical delivery, pre-formulation, solubility, saturation, skin penetration, chemical stability, rheology, excipients, analytics

## Abstract

Specific aspects of semi-solid dosage forms for topical application include the nature of the barrier to be overcome, aspects of susceptibility to physical and chemical instability, and a greater influence of sensory perception. Advances in understanding the driving forces of skin penetration as well as the design principles and inner structure of formulations, provide a good basis for the more rational design of such dosage forms, which still often follow more traditional design approaches. This review analyses the opportunities and constraints of rational formulation design approaches in the industrial development of new topical drugs. As the selection of drug candidates with favorable physicochemical properties increases the speed and probability of success, models for drug selection based on theoretical and experimental approaches are discussed. This paper reviews how progress in the scientific understanding of mechanisms and vehicle-influence of skin penetration can be used for rational formulation design. The characterization of semi-solid formulations is discussed with a special focus on modern rheological approaches and analytical methods for investigating and optimizing the chemical stability of active ingredients in consideration of applicable guidelines. In conclusion, the combination of a good understanding of scientific principles combined with early consideration of regulatory requirements for product quality are enablers for the successful development of innovative and robust semi-solid formulations for topical application.

## 1. Introduction

Most topically applied dosage forms are semi-solid. Regarded as a physical body, they combine the properties of liquids and solids. At rest, they behave more like a solid, which, for example, allows a prolonged residence time at the site of application and stability against sedimentation. Under shear, they liquify and can be spread gently and evenly on the skin surface [1]. As one of the oldest dosage forms, they still play an important role in medicine, especially in dermatology, but also for nasal or ophthalmic delivery, treatment of wounds, or delivery on the oral, vaginal or rectal mucosa. Topical delivery is attractive as it allows to bring the active directly to a localized region of the body without exposing the whole organism systemically, thus reducing issues of toxicity and side effects [2]. Moreover, as opposed to peroral administration, issues of degradation in the gastro-intestinal tract or first-pass metabolism can be avoided.

More than many other dosage forms, topical formulations can directly influence the wellbeing of the patient. They may provide instant relief–or discomfort. Effects on the skin, like hydration and trans-epithelial water loss, may persist for several hours. On the other hand, the onset of clinical effects is often slow, and therefore, an aesthetically pleasing and well-tolerated formulation is an important contribution to compliance and clinical success [3].

In the past decades, advances have been made in the understanding of topical drug delivery. This includes knowledge of the skin barrier function and driving forces for skin penetration [4,5], the influence of solvents [6,7] and formulation parameters that influence skin penetration [8], advances in spectroscopic tools to investigate skin penetration [9,10], improved instruments and approaches for assessing physical stability of formulations [11], and a better understanding of the role of excipients in topical drug delivery [3]. Still, in terms of formulation types, traditional formats like ointments and creams prevail, and the nomenclature of the formats is more historically based than technically justified [12]. Furthermore, since systematically developed synthetic emulsifiers and thickening polymers of synthetic or partially synthetic origin became readily available more than 50 years ago, the types of delivery systems for approved topical drugs have not changed significantly. This is opposed to the situation in per oral drug delivery, were an increasing number of novel formulation options like solid dispersions, microemulsions, and nanoparticulate systems have been approved during the last 30 years. Similarly, liposomal or solid-lipid nanoparticle-based formulations have been introduced as novel drug delivery systems for injectable drugs. In academic research, several studies on novel approaches like nanocarriers such as solid lipid nanoparticles (SLN) or nanostructured lipid carriers (NLC) for topical delivery have been published [13,14]. It has been demonstrated that nano-particles of specific size ranges can penetrate into the hair follicle; however, certain massage procedures are necessary for this [15,16], which limits the applicability of the approach. In combination with the challenging aspects of the long-term stability of such formulations, they have not gained industrial importance in medicinal dermatology.

In this article, we outline how the progress in understanding topical drug delivery and advances in technology for formulation characterization can enable a more structured and rational design approach to topical drug products. The focus is on early development activities such as formulation design and prototype development and on medicinal products from an industry perspective. In contrast to academic studies, which are typically focused on performance aspects of formulations, in an industrial context, further aspects need to be considered. Most importantly, this is product quality. Quality requirements are described in the guidelines of the “International Council for Harmonisation of Technical Requirements for Pharmaceuticals for Human Use” (ICH) [17] and are largely harmonized between the USA, Europe and Japan. In addition, the respective Pharmacopeias may provide further specific guidance. The individual guidelines and their specific requirements should not be discussed within the scope of this article, but their consequences for product development shall be briefly outlined. Quality, according to ICH Q6, is “the suitability of either a drug substance or drug product for its intended use. This term includes such attributes as identity, strength, and purity” [18]. Important quality attributes need to be specified for release and for the end of shelf life. Therefore, an important aspect of quality is the stability of the drug product. The shelf life is defined as “the time period during which a drug product is expected to remain within the approved shelf-life specification, provided that it is stored under the conditions defined on the container label.” Typically, a shelf life of at least two years at room temperature is required for reasons of commercial viability for topical drug products (the exact conditions for stability testing depend on the respective climate zone as per WHO definitions [19]). One attribute that is always specified is the assay of the active ingredient(s) (API). The standard limit is 95–105% of the declared concentration in Europe [20] and 90–110% in the US [21]. Specifications for physical stability are derived from critical quality attributes (CQA) and may include pH, viscosity, and particle or globule size distribution for semi-solid products. As it will be explained in more detail in the section on analytics below, it takes about four months to arrive at an initial robust stability forecast on chemical stability based on accelerated studies. Although some tools exist to obtain indications on long-term physical stability, it is often even more challenging to predict it and it may be that real-time data are necessary for a definitive confirmation. As APIs are typically present in dissolved form, the risk of degradation is increased as compared to solid forms. As semi-solid formulation are often thermodynamically instable, stability is a particular concern. As a consequence, product stability is a major risk factor in industrial development as inaccurate predictions and failure to achieve stability are very costly and time consuming and, therefore, have a strong impact on the commercial viability of a development project. Other constraints to be considered in industrial development are time to market, regulatory compliance, in particular with regards to the quality and approval status of excipients. Furthermore, attention is required on primary packaging options, cost of goods (excipients, packaging and manufacturing process) and the options for scale up and industrialization of the manufacturing process. In addition, aspects of intellectual property need to be considered. For a new chemical entity (NCE) with IP protection, typically ‘freedom to operate’ is critical as an infringement of existing patents would delay development timelines and/or increase costs. For repurposing or reformulation of already existing APIs, often a patent protection of the formulation is desired. Overall, the industrial development of novel semi-solid formulations is a process where, in addition to scientific and product performance aspects, many other constraints need to be considered. In particular this is product quality which has a major impact on development cost and likelihood of success.

In this work, we briefly describe and discuss the status quo of industrial development of semi-solid formulations based on a theoretical analysis of the semi-solid dermatological NCE formulations approved in the last seven years. We then discuss candidate selection and propose guiding principles of science-based, rational design of such formulations as well as the importance of understanding excipients. Lastly, some central approaches to characterize semi-solid formulations with respect to important quality attributes are presented. The focus is on formulations for topical delivery into the skin, although many aspects will also apply to other topical delivery of semi-solid dosage forms, e.g., for ocular, mucosal, or transdermal delivery.

## 2. Status-Quo of Semi-Solid Formulations in Recently Approved Products

In order to get insights into the current status of the development of semi-solid formulations of medicinal products, it is relevant to analyze the products which actually entered the market. As opposed to formulations only described in the scientific literature or patents, formulations of products that have been approved and commercialized must have fulfilled the above-mentioned quality and stability criteria and the further constraints of industrial development.

In Table 1, the qualitative composition of the semi-solid formulations for NCEs approved by the FDA between 2016 and 2022 are summarized [22]. Five are creams and two ointments. The data provide relevant insights; however, as the quantitative composition is typically not disclosed, only conclusions can be drawn on the underlying development strategy. Moreover, pre-formulation data from which essential parameters of rational formulation design such as solubility, saturation and distribution of the API could be derived are typically not available.

The Eucrisa^®^ appears to consist of a ‘classical’ hydrocarbon ointment base with mono- and di-glycerides as consistency agents and propylene glycol as a solvent. The design principle is similar to the Protopic^®^ ointment, where propylene carbonate instead of propylene glycol is dispersed as a solvent for the API in the ointment matrix in the form of an “oil-in-oil” emulsion [23].

The reasons to choose this approach may be solubility (e.g., a strong drop in solubility if the solvent propylene glycol is mixed with water) or to avoid hydrolytic degradation of the API. Petrolatum-based ointments are generally occlusive, which may support API penetration but are also characterized by greasiness which is not always preferred [24,25]. Klisyri^®^ ointment consists of two excipients only. For propylene glycol, the concentration of 89% (*w*/*w*) is disclosed in the summary of product characteristics by the European Medicines Agency (EMA) [26]. As the active tirbanibulin is contained at 1% (*w*/*w*), it can be concluded that mono- and di-glycerides are present at 10% (*w*/*w*), so the full quantitative composition is available in this case. Propylene glycol acts as a solvent, and mono- and di-glycerides are used to thicken the formulation to an ointment-like consistency. Technically, the formulation may be better described as an alcoholic gel. Such high concentrations of propylene glycol are typically not preferred for tolerability, skin feel, and cosmetic appeal. However, Klisyri^®^ is approved for the 5 day treatment of actinic keratosis, which typically affects only a small area of the body surface. For this indication, the formulation type appears acceptable.

The five creams can be assumed to be oil-in-water (O/W) emulsions based on the type of emulsifiers used. As an oil phase, hydrocarbons (Winlevi^®^), triglycerides (VTAMA^®^), or mixtures of triglycerides with silicones (Aklief^®^), mixtures of fatty acid esters with hydrocarbons (ZORYVE^®^) or mixtures of hydrocarbons, triglycerides and silicones (Opzelura^®^) have been used. All formulations contain a hydrophilic co-solvent. In one case, this is propylene glycol alone; in two cases, this is propylene glycol in combination with another hydrophilic co-solvent; and in one case, this is a combination of hexylene glycol with diethylene glycol monoethyl ether. These hydrophilic solvents can have several functions in a formulation: they can act as solvents for the API, as penetration enhancers [27], and can contribute to the preservation of a formulation. Their general role as penetration enhancers will be discussed below in the section on formulation design. With regards to the physicochemical properties of the APIs formulated as creams, in four cases, the active is present in the neutral form with calculated logP values between 3.7 for clascoterone and 5.2 for trifarotene according to drugbase.ca and, therefore, in the clearly lipophilic range for which low water solubility can be expected. These APIs may have good solubility in the above-mentioned solvents, but a significant drop in solubility can be expected when mixed with water. In contrast, decent to good solubility in polar oils such as medium-chain triglycerides may be possible, whereas hydrocarbons and silicones show poor solubility for almost all APIs. Therefore, it appears that the solubilization systems were not fully optimized for all of the creams. A more detailed discussion on solubilization will follow below in the section on formulation design. In only one product, Opzelura^®^, the API is present as salt (ruxolitinib phosphate). It can be assumed that the reason for using the salt is that this form is also used in the previously approved peroral tablet, Jakafi [22]. Ruxolitinib phosphate is reported to have a good water solubility of around 1% at acidic and slightly acid pH values [26]; therefore, it can be assumed that it is predominantly dissolved in the water phase. The presence of the hydrophilic co-solvents polyethylene glycol 200 and propylene glycol in the water phase may further increase the solubility of the active. Overall, Opzelura^®^ is a very complex formulation with 16 excipients which probably could have been designed in a much leaner way in light of the assumed good solubility of the active in the aqueous phase.

In O/W creams, stabilization of the oil droplets in the continuous water phase is typically obtained by emulsifiers or, alternatively, by polymers with surface-active functionality. Semi-solid consistency is obtained by the addition of thickening polymers or constancy agents such as fatty alcohols or mono- and diglycerides. In Winlevi^®^, VTAMA, and ZORYFE, a mixture of three to four hydrophilic emulsifiers and lipophilic co-emulsifiers/consistency agents is used, and in Opzelura^®^ is used a mixture of five. In such systems, hydrophilic emulsifiers and consistency agents form liquid crystalline structures, which build the systems’ viscoelastic properties. In Aklief^®^, in contrast, a copolymer of acrylamide and acryloyldimethyl taurate is used to stabilize the oil droplets due to its surface-active functionality. It also acts as a thickener to provide a semi-solid structure. In such a case, no surfactant-type emulsifier is necessary, which can improve the tolerability of a formulation on sensitive skin. Topical formulations, especially when containing water, need to demonstrate antimicrobial efficacy. For this, either alcohols, typically in concentrations of more than ten percent, or preservatives, or a combination of both need to be present. From the absence of a preservative in Winlevi^®^, it can be concluded that a substantial concentration of propylene glycol was used. The antioxidants used may be necessary to stabilize the API or other ingredients. Buffers and pH-modifiers are added to guarantee pH values needed for chemical stability or to optimize skin tolerability.

Apart from the type of emulsifiers used, the Winlevi^®^, VTAMA^®^, Opzelura^®^ and ZORYFE^®^ creams show a general resemblance to established pharmacopeial cream compositions [25,28]. A characteristic feature of classical creams is that semi-solid consistency is achieved by liquid crystalline phases formed from hydrophilic emulsifiers in combination with consistency agents, as described above. This requires a concentration of hydrophilic emulsifiers of several percent, which may not be ideal for tolerability on sensitive skin. In addition, for consistency agent stabilized formulation, process parameters often have a strong impact on product quality [29,30], which can make scaleup cumbersome. It has been shown that creams based on polymeric thickeners can be stabilized with less than 0.2% hydrophilic emulsifiers and can be better understood in terms of API saturation and distribution [31,32]. Therefore, apart from cases where the specific sensorial properties of consistency agent stabilized creams are desired, polymer-thickened alternatives can be typically more rationally designed, upscaled, and controlled.

In conclusion, based on the available information, it appears that recently approved novel semi-solid formulations are still frequently based on traditional formulation approaches. However, systematic analysis is difficult because pre-formulation and formulation work on commercial formulations is often not published. Even when quantitative compositions are reported in patents, the design principles and justification for the chosen approach are typically not provided. It is not possible to determine whether this lack of information is due to the fact that the development work was performed using an empirical rather than a systematic approach or whether it was omitted for other reasons. In general, we would encourage the sharing of more scientific information on the formulation design of marketed products wherever possible for intellectual property reasons. Transparency in formulation design can increase confidence in medicines and contribute to overall scientific progress in the field.

## 3. Formulation Design

### 3.1. Selection and Formulatability Assessment of Active Ingredients

New chemical entities for topical drug delivery are often not originally optimized for topical application from a medicinal chemistry point of view. Rather, they frequently come from peroral development programs and are then shifted to topical development. Therefore, their physicochemical properties are often not ideal for skin penetration and topical formulatabilitiy. However, selecting the right molecule from the candidates of a development program offers great potential. A well-suited molecule increases the likelihood that a technically robust, aesthetically pleasing, and well-tolerable formulation is developed and that development timelines are shorter, and the risk of failure is lower. Furthermore, the API concentration may be reduced, which saves costs, improves safety with regard to application on compromised skin or through contamination, and reduces the impact on the environment.

For theoretical compound selection, two major aspects need to be considered: the expected ability of the candidate to penetrate into the skin and its technical formulatability. With regard to estimating skin penetration from the physicochemical parameters of a drug, a number of rules have been proposed in the literature. It is generally acknowledged, however, that Lipinski’s Rule of Five [33] does not apply well to topical compounds. One reason for this is the much lower permeability of the stratum corneum as compared to intestinal epithelial cells [34].

A popular assumption was that molecules have to have a logP value from one to three and a molecular weight below 500 [25]. While this may be largely true for transdermal delivery, criteria for dermal delivery (i.e., target within the skin) are more complex, and many drugs approved in the past 20 years are outside this range (e.g., tacrolimus: logP = 3.3, MW = 804 Da, ivermectin: logP = 5.8, MW = 875 Da, bexarotene: logP = 6.9, MW = 349 Da).

In general, it can be assumed that low molecular weight is advantageous [35,36]. For a first estimate of skin permeability based on basis physicochemical properties, often the Potts-Guy equation (Log (C_perm_) = −6.3 + 0.71 × log P − 0.0061 × MW) is used [37]. The prediction was obtained from a fit to a dataset of around 90 compounds with logP values ranging from −3 to +6 and molecular weights ranging from 18 to >750 Da. In addition, a low product of H-bond donors and acceptors is favorable [38]. Furthermore, the dipole moment/symmetric distribution of the hydrophobic/hydrophilic moieties of the molecule is favorable [39]. Lastly, owing to the hydrophobic nature of the stratum corneum as the major barrier to penetration, charged molecules penetrate significantly less than neutral ones. For ionized proton bases, a factor of 15–20 times lower permeation has been demonstrated [36]. For anions, the permeation is even lower than for cations due to the presence of free fatty acids in the stratum corneum lipid mixture, and there is an additional electrostatic repulsion.

For technical formulatability, two approaches to estimate formulatability based on the physicochemical properties of the compound may be considered. A general approach by Faller and Ertl from Novartis [38] is based on comparing the logP and the logarithmic inverse molar water solubility of a compound; it can be analyzed whether the solubility is mainly limited by lipophilicity or if additional factors are responsible for poor solubility. This approach was initially developed with a focus on peroral drug delivery, but there are strong indications that it also applies to topical formulation development (M. Herbig, unpublished data). Here, it is important to use the (intrinsic) thermodynamic water solubility data and not data generated from buffers. The most important of these additional factors is high lattice energy which makes dissolution energetically more costly. This can be expressed as “Delta Log 1/WS-LogP”; the higher it is, the more challenging it is in general to formulate a compound.

Santos et al. [40] based their model on data of 114 approved topical molecules. Four parameters that correlate with good developability for topical dermatology have been identified: (i) topological polar surface area (TPSA) ≤ 100 Å^2^, molecular weight ≤ 500 Da, calculated logarithmic partitioning coefficient (cLogP) = 1.0 to 4.0, and aromatic ring count ≤ 2. Of the approved topical compounds, 71% show zero or one deviation from these criteria, 22% shows two, and only 7% shows three or more. However, it needs to be taken into account that this is an empirical relationship. In recent decades, API got generally larger and more lipophilic [38], and challenges in permeability and formulatability have been at least partially compensated for by the higher potency.

Wherever possible, an experimental candidate screening should be performed. An approach that can be performed with less than 100 mg substance may be a truncated solubility screening in a limited number of aqueous and non-aqueous solvents, preliminary forced degradation studies in order to assess the intrinsic stability profile, and an investigation of skin penetration from simple solvents. If a higher number of compounds or vehicles should be screened, and if appropriate controls are applied, the latter can also be performed as cassette dosing. This means that several compounds are dissolved in the same vehicle in order to save resources and time.

Overall, whenever several candidates for development are available, always a systematic compound selection should be performed. For very high numbers of compounds or if no compound is available for screening purposes, this may be performed at a theoretical level which, however, will have a more limited predictive value. Wherever some compound in the double-digit milligram range can be made available, an experimental screening should be performed. Selecting the right compounds can greatly increase the likelihood of success of a topical development program and reduce cost and time.

### 3.2. Rational Formulation Design Approaches

The rational design of formulations is API-centric. It is based on a comprehensive characterization of the physicochemical properties of the API. The formulation vehicle will be designed to best accommodate the requirements of the API. In contrast, a vehicle-centric approach incorporating APIs in different vehicles and testing them for stability and performance will not allow a systematic understanding of the formulation and bear the risk that development work needs to be started from scratch if issues arise.

A prerequisite of a systematic and rational formulation design is a good understanding of the driving forces for a molecule’s skin penetration, including the influence of the vehicle. In a review of rational cosmetic formulation design, Lane et al. [8] provided an excellent overview of approaches to rationalize the understanding of skin penetration under consideration of vehicle effects. From the Higuchi equation, it can be derived that the concentration gradient and the thermodynamic activity of the active are major driving forces for skin penetration. In a formulation, the thermodynamic activity can be optimized by providing a high degree of saturation. However, care must be taken that the degree of saturation during the shelf life of a formulation does not exceed 100%; otherwise, the API may precipitate, which is an unacceptable product change from a quality point of view. However, if supersaturation of the active occurs after application on the skin (e.g., through the evaporation of a solvent), it may further enhance penetration [41]. Lane et al. also discuss the concept of “formulating for efficacy,” as introduced by Wiechers et al. [42] and further developed by the use of Hanson solubility parameters by Abbott [43]. In brief, formulation development by optimization of the relative polarity of actives, vehicles, and stratum corneum is proposed. Ideally, the absolute solubility of the active in the vehicle should be high, but it should be lower relative to the stratum corneum solubility in order to promote penetration. Although the general considerations behind the approach are valuable and may be considered in formulation development, no universally applicable algorithm for formulation development can be derived from the model [8].

Overall, saturation of the active in the formulation is often the major driving force for skin penetration. In order to determine saturation, comprehensive thermodynamic solubility data in solvents and solvent combinations need to be generated. For tacrolimus, a calcineurin inhibitor used for the topical treatment of atopic eczema as Protopic^®^ ointment, data on solubility, saturation and release and/or penetration are available in the literature. The approved ointment is a droplet dispersion-type ointment with 5% propylene carbonate dispersed in a petrolatum-based matrix. It represents one of the few examples of a marketed formulation for which a detailed discussion of the formulation design and release kinetics is described in the literature [23]. Tacrolimus is dissolved at low saturation in the propylene carbonate droplets. Upon application, propylene carbonate starts to diffuse from the ointment matrix leading to an increase in saturation and, ultimately, the amorphous precipitation of tacrolimus. As long as tacrolimus is present in the dissolved state, the release rate is low. Only as the active has been precipitated does the thermodynamic activity (saturation is 100%) and the release rate increase. Such a release mechanism may support a prolonged release of drugs and limit systemic exposure.

Developing tacrolimus as a cream formulation is challenging for the instability of the molecule in the presence of water and for matching the skin penetration profile to Protopic^®^ ointment. Interestingly, most cream formulations showed higher skin penetration than the ointment [44], which, however, is undesired for potential systemic side effects of tacrolimus. In a recently published patent [32], solubility data of tacrolimus in different lipophilic solvents (oil phases) have been shown. The values range from 0.007 mg/mL in liquid paraffin to around 63 mg/mL in diisopropyl adipate, a span of almost four orders of magnitude (Table 2). As solubility in different oil phases can be so different, it should be avoided to talk about “the oil solubility.” In particular, hydrocarbon solvents such as paraffin show very poor solubility for most APIs and typically can be considered antisolvents. As shown by solubility data from the combination of different oils, mixing oils allows targeted adjustment of solubility and saturation. By decreasing the degree of saturation, an initially too-high skin penetration profile could be well-matched to that of the Protopic^®^ ointment [32]. For betamethasone dipropionate, a decrease in saturation during the transformation of the vehicle led to lower skin penetration as compared to formulations where the saturation did not significantly change [31].

A further important aspect to be considered in formulation design is the transformation of the formulation vehicle after application on the skin, which sometimes is also termed metamorphosis [12]. As water or other volatile solvents evaporate, the nature of the formulation vehicle may change significantly. The saturation may increase if it is a solvent that evaporates or decrease if an antisolvent evaporates. Especially in the case of topical film-forming solutions (FFS), it has been described that supersaturation occurs during solvent evaporation which enhanced release of beclomethasone dipropionate [45] or the permeation through pig ear epidermis of a lipophilic experimental drug [46]. In addition to supersaturation, also the nature of the resulting film can help to improve and prolong the release profile, as demonstrated for different combinations of polymers and plasticizers [47]. Recently, it was demonstrated that the principle by which peroral solid dispersions can enhance bioavailability could also be applied to topical FFS. When the resulting films represent true solid solutions of the API in the polymer, superior release and skin penetration as compared to films without molecular miscibility of drug and polymer could be demonstrated [48]. As compared to a reference cream formulation, the penetration enhancement was even more pronounced. As conventional formulations are easily removed from the skin by skin-to-skin or clothing-to-skin contact [49], topical FFS can also contribute to product safety, especially for highly potent drugs used for localized treatment.

A further aspect of importance for the understanding of topical formulations and their rational design is the understanding of partitioning or distribution between different phases of a formulation, most importantly, between the water and oil phases of emulsion-type systems. As described above and shown in Table 2, the solubility of an API in various oil phases can be very different. Similarly, also the partition of a molecule between different oil phases and the water phase can be very different, with consequences on the performance of the formulation. As shown in Figure 1, the logP of the frequently used preservative phenoxyethanol between liquid paraffin and the buffer is around −0.7, which means that around 80% of the preservatives are present in the water and around 20% in the oil phase. The logP between MCT and buffer, however, is around +0.7, which means the distribution pattern is reversed, and only around 20% of the substance is in the water phase. As it has been shown that the preservation efficacy of a formulation correlates rather with the free concentration of the preservative in the water phase than with its nominal concentration in the formulation [50], a good match between the preservative and oil phase is important for efficient product development. With the addition of increasing concentrations of the hydrophilic solvent PEG400 to the water phase, the distribution from MCT to water increases (Figure 1). For hydrolysis-prone APIs, a strong partitioning into the oil phase is a physical means of improving chemical stability, as shown for betamethasone dipropionate and tacrolimus [31,32]. Apart from the design of the oil phase, also minimizing the excess concentration of the hydrophilic emulsifier plays an important role in this.

Apart from optimizing saturation and transformation of a formulation, also chemical penetration enhancers can be a tool for improving skin penetration. Lane [27] provided a comprehensive overview of penetration enhancers used in topical drug delivery. There are two main mechanisms of action, lipid chain interaction of penetration enhancers which leads to the fluidization and increased permeability of the stratum corneum, and penetration enhancers, which increase the solubility and partitioning of actives in the stratum corneum. However, there is no universal penetration enhancer. Penetration enhancement needs to be regarded on the level of the whole formulation, taking into account solubility, saturation, partitioning and transformation. For example, propylene glycol, the most commonly used penetration enhancer, has even been shown to decrease penetration as compared to a number of other solvents that are not considered penetration enhancers [44].

It has also been discussed whether skin penetration enhancement can be achieved by the use of liposomal formulation approaches. It has been found that liposomes can both enhance or reduce the penetration of actives into the skin, dependent on the properties of both the API and the lipids. Elastic vehicles generally have superior properties as compared to rigid vehicles [51,52]. Although occasionally disputed, there is general evidence that intact liposomal vehicles do not cross the stratum corneum [51,53]. The long-term physical and chemical stability of liposomes in the form of a semi-solid formulation is challenging; therefore, they have not gained importance in topical drug delivery.

In the context of rational formulation design, it is worthwhile to discuss the role of “Quality by Design (QbD)” in topical formulation development. QbD has been introduced as a formal framework in drug development described in the ICH guidelines Q8, Q9 and Q10 [17]. According to ICH Q8 [54], an objective of QbD is “to provide a comprehensive understanding of the product and manufacturing process,” and it is often contrasted by a traditional approach of “quality by testing (QbT)” [55]. Important aspects of the QbD framework are the definition of a quality target product profile (QTPP), a comprehensive understanding of critical material attributes (CMA), critical process parameters (CPP), and the definition of critical quality attributes (CQA) and a design space for the resulting product. Formal experimental design or “Design of Experiment (DoE)” studies are suggested for product optimization. The scope of QbD is Module 3 of the Common Technical Document (CTD) submitted for marketing authorization applications. This does not apply to submission during the clinical research stage, although the use of its principles during this phase is encouraged. The role of the ICH QbD framework in formulation development in general and in topical formulation development, in particular, has been previously discussed [55,56,57] and shall not be the focus of this review.

Beyond this formal framework, Quality by Design can also be understood as a more general approach and ambition in formulation development based on a commitment to systematically understand the formulation, which is very similar to what we understand as “rational design.” From the author’s point of view, it is highly recommended to apply general “Quality by Design” or “Rational Design” principles from the very beginning of formulation development. In most cases, it is advisable to start with a preliminary QTPP that is refined based on the knowledge gained during exploratory development. The data generated, e.g., on solubility and saturation, will provide an important basis for the formal establishment of formal QbD principles during clinical/confirmatory development. DoE approaches are particularly useful in confirmatory development, e.g., for the optimization of antioxidant systems, emulsifier systems, or manufacturing process parameters.

In general, the authors believe that the following four aspects of a formulation as a topical drug delivery system are particularly important to investigate and understand in order to enable rational formulation design: (i) the solubility of the API(s) in the formulation, (ii) the degree of saturation, (iii) the distribution processes within the formulation, and (iv) considerations about the transformation of the formulation after application. A good understanding of these parameters will increase the likelihood of success and reduce development time in current formulation development projects.

### 3.3. Excipient Characterization

Many excipients used for topical drug delivery are complex. They are often mixtures, some of natural or semi-synthetic origin, often with broad specifications. Batch-to-batch variability within the specification may, therefore, influence critical quality attributes of a formulation. In the following, some examples of excipient variability and its impact on topical formulations are highlighted.

Petrolatum, still the most commonly used ointment base, can be considered a formulation in itself. It is an oleogel composed of two solid hydrocarbon waxes and a liquid hydrocarbon fraction. The specifications of the European Pharmacopeia (EP) with regards to its physical properties (consistency by penetrometry and drop point) are wide and without direct relevance to typical critical material attributes in topical formulations. An investigation of 14 commercially available petrolatum variants in EP quality revealed a strong variation variability in rheological properties [3]. However, rheological properties may be critical for different performance attributes: the zero-shear viscosity can affect diffusion and, thus, the release and penetration of APIs, and viscosity at higher shear rates affects spreadability. The complex modulus is a measure of the rigidity of the petrolatum variant, and the flow point correlates with its structural strength–both values show variability by almost one order of magnitude (Figure 2). Therefore, different variants are suitable for different purposes, and the exchangeability of variants affecting product quality is generally not guaranteed.

Glycerol monostearate 40–55, type 1, according to EP, contains between 16% and 33% glycerol monostearate; the remainder of the species are glycerol monopalmitate as well as di- and triglycerides. Depending on the composition, the emulsifying and consistency-increasing properties, as well as the solubilizing potential for APIs, may vary. Similarly, medium-chain triglycerides can considerably vary in fatty acid composition and distribution pattern of fatty acids in the triglyceride ester with potential batch-to-batch variability with regards to the solubility of different APIs.

Perhaps the most complex excipients are polysorbates. For polysorbate 20, around 700 different subspecies over a wide range of lipophilicity have been identified [58]. This is particularly important with regard to the stabilization and physical stability of injectable protein formulation, but variation in the composition can also impact solubilization or emulsification efficacy in topical formulations.

In pharmaceutical emulsions, often fatty acid ethers are used as emulsifiers for their efficacy and chemical stability [1]. A particularly interesting variant is lauromacrogol 400, also known as polidocanol, which, apart from its use as an excipient, may also be used as an active ingredient for local anesthetics or for vein sclerotherapy. According to the European Pharmacopeia monograph for the drug substance [59], the number of moles of ethylene oxide reacted per mole of lauryl alcohol is nine. However, as shown by data generated by UPLC-MS analysis (Figure 3), it could be demonstrated that the number of ethylene oxides moles per mole lauryl alcohol is 10–11 and that differences between suppliers exist (Evers DH, unpublished data).

Whenever such excipients with high variability in specifications are used in formulations, it is important to assess which properties are critical for the performance of the drug delivery system. Ideally, a design space covering the variability can be established. If this is not possible, internal sub-specification may need to be established.

The authors are convinced that it is important to identify critical material attributes (CMA) in formulations and to characterize the corresponding materials appropriately. When an excipient is identified as critical to the quality of the resulting product, it is important to consider the potential impact of excipient variability between manufacturers or from batch to batch. Novel UPLC-MS methods allow the characterization of even very complex surfactant-type excipients, and modern rheology allows a better understanding of semi-solid excipients such as petrolatum. Overall, a solid understanding of critical excipients is an important aspect of risk mitigation in topical formulations.

## 4. Characterization of Semi-Solid Formulations

Topical semi-solid formulations, especially when emulsion-based, are often systems of high complexity, requiring sophisticated approaches for the characterization of chemical, physical, and microbial stability. In earlier stages of development, characterization is more focused on decision-making, knowledge-gaining, and risk mitigation, whereas the focus in later stages is on establishing data packages for submission and quality control. Especially in the context of generic drug development, the microscale organization of matter in semi-solids is often described as “microstructure.” Physical characterization may include microscopy, rheology, light scattering and laser diffraction techniques for particle size characterization, and a number of spectroscopic techniques. For comprehensive overviews of microstructure characterization in topicals, the reader can be referred to the book by Langley, Michniak–Kohn and Osborn [60] and the article by Badruddoza et al. [11]. What will be discussed in the following are applications of rheology for formulation characterization, an area that gained importance during the last years and were interesting case studies became available. Approaches to address chemical stability in semi-solid formulations are less described in the literature than aspects of physical stability but are often critical for successful development. Therefore, also this aspect will be discussed. Finally, a brief overview of in vitro methods to investigate release, penetration, and permeation form semi-solid formulations will be given.

### 4.1. Rheological Characterization of Semi-Solids

The rheological characterization of semi-solid formulations provides an important source of information on various aspects relevant to the systematic development and characterization of semi-solid formulations. Rheological characteristics can be used to describe several technological processes, such as pipe flow, properties during stirring or spraying, product removal from the packaging or how the product can be applied on the skin. Semi-solid formulations are typically structured liquids that follow non-Newtonian flow behaviors. This means that their viscosity is dependent on the shear applied. Most semi-solid dosage forms show a shear-thinning flow behavior. Whereas the viscosity at moderate and elevated shear rates is easier to determine, viscosity values at very low shear rats of 0.001 1/s or below can only be measured by modern high-performance rheometers. Values at such low shear rates are often also referred to as ‘zero shear viscosity.’ Whenever the viscosity of a formulation at rest needs to be described, the zero-shear viscosity needs to be considered. This refers to the phenomena of sedimentation or creaming in emulsions and suspensions. According to the Stokes equation, the speed of creaming or sedimentation is inversely proportional to the dynamic viscosity [1]. According to the Stokes–Einstein equation, also the diffusion coefficient is inversely proportional to the viscosity. Whenever systems at rest are described, the zero-shear viscosity needs to be considered. In Figure 4, the viscosity curves of two creams thickened by different polymers are presented in double-logarithmic presentation (M. Köllmer, unpublished data). Whereas the viscosity at high shear rates and, therefore, the spreadability is almost identical, cream 1 has a five-fold higher zero-shear viscosity which makes it more stable against creaming or sedimentation.

The viscoelastic properties of semi-solid formulations can be obtained using oscillatory tests such as the amplitude sweep or the frequency sweep test. The most relevant parameters for semi-solid formulation development include the loss modulus (symbol G″, unit Pa) that describes the viscous content of a formulation, which is a measure of the lost deformation energy during the test. This energy is used for the alteration of the sample structure and is delivered to the surrounding area. Ideal viscous substances have the same loss moduli before and after stress application. The storage modulus (symbol G′, unit Pa) describes the elastic content of a formulation. It is a measure of the stored deformation energy during the test. Ideal elastic substances have the same storage moduli before and after stress application. The phase angle (symbol δ, unit °) describes the ratio between viscous (liquid-like) and elastic (solid-like) components. A value of 0° describes an ideally elastic material (e.g., steel), and a value of 90° describes a liquid such as water. Materials with phase angles < 45° do not flow when at rest. For comparable materials, a lower phase angle is an indication of a more pronounced internal structure. The flow point (symbol τ_FP_, unit Pa) is the shear stress value at the intersection of the curves of G″ and G′. If the flow point is exceeded, the viscous portion dominates over the elastic part of a sample, and it flows. A high flow point often is a measure of good structural stability.

Rheological investigations can contribute to the characterization, prediction and optimization aspects of formulation aging and stability [61,62]. Changes in the formulation microstructure over time or after exposure to different temperatures can often be detected more sensitively in the rheological profile than in macroscopic or microscopic examination. In particular, the rheological swing test—also called the temperature cycling test—introduced by Brummer [62], can provide valuable contributions to the prediction of the long-term physical stability of emulsion systems. If the loss factor tanδ (or, alternatively, the storage or complex modulus) remains constant after several temperature cycles, this is a good indicator of long-term physical stability. Formulations that show significant changes in these parameters, in contrast, are prone to physical instability.

Furthermore, inter-batch variability due to different excipient batches, altered manufacturing conditions, or during storage can be determined. The similarity of the microstructure of a generic cream to an originator product can be assessed by rheological equivalence testing. The EMA Draft guideline on the quality and equivalence of topical products from 2019 provides specific guidance on equivalence testing for topical products in lieu of clinical equivalence trials [63].

Rheological investigations have also been used to investigate correlations of viscosity with release and permeation from semi-solid formulations. Whereas, as expected from the Stokes-Einstein equation, a negative correlation of viscosity with the release rate could be demonstrated, no clear correlation with skin permeation or penetration could be observed [64,65]. The latter is not surprising, as the rate-limiting step for penetration into the skin is rather the diffusion through the stratum corneum than the diffusion within the formulations. Furthermore, rheology has proven to be useful in predicting certain sensorial properties [66,67] and process parameters. In a recent study, it has been demonstrated that a thixotropic emulsion gel-based nasal spray could be optimized by means of rheology for device compatibility and improvement of nasal residence time [68].

In the opinion of the authors, a comprehensive rheological analysis should be part of the formulation development and optimization work for semi-solid formulations. However, it is not recommended to perform the same set of tests for each formulation but to select the rheological test based on an analysis of which parameters are critical to the performance of a particular formulation. For example, for a pourable lotion, zero-shear viscosity may be critical because it correlates with stability against creaming or sedimentation if it contains suspended particles. In contrast, for a petrolatum-based ointment, it may be important that the viscosity remains below a certain limit to ensure acceptable spreadability on the skin.

### 4.2. Chemical Analytics of Semi-Solids

The development of analytical methods is equally important as formulation development. Successful product development consists of the close interlinking and collaboration of formulation development and analytical method development. Only with efficient and reliable analytical methods can product quality be ensured, optimized and controlled. In other words, the certainty of having a good product can only be as high as the quality of the methods used to analyze it. In this section, we focus on methods for chemical analysis, which involve methods to determine the content of the active ingredient and its degradation products and, if applicable, the assay of functional ingredients such as preservatives or antioxidants. Most commonly, high-performance liquid chromatography (HPLC) is used for that purpose [69]. Typically, a reversed-phase column and UV detection are used [70].

In many cases, analytical method development for semi-solid formulations is more challenging, for example, for solid dosage forms. One reason is the typically unfavorable drug-to-matrix ratio. A tablet often consists of around 10–50% active ingredient, whereas for semi-solid formulations, this is often between 0.1% and 1%. The level at which degradation products need to be determined can be calculated from their concentration in the formulation and the (anticipated) maximum daily dose (MDD) according to ICH Q3B [71]. An illustration of the levels of the reporting, identification, and qualification thresholds for a 0.1% formulation dependent on the MDD is shown for a formulation with 0.1% API in Figure 5. It can be seen that impurities often need to be determined at a 0.1% level of the active ingredient. For a product with 0.1% API, this results in a quantification limit of 0.0001% for the impurity. Apart from the need for well-developed methods and sensitive equipment, this may also need sophisticated extraction procedures from the formulation matrix. The task of the formulator is to develop a stable formulation that does not easily disintegrate upon thermal or mechanical stress, whereas the analytical scientist has to extract both the API and its impurities quantitatively from the same formulation matrix. Creams and ointments are often composed of excipients with different polarity, solubility, and melting points. A good understanding of chemistry, formulation design, excipient properties and often creative problem-solving skills are necessary to develop efficient and robust extraction procedures. Another specific challenge of semi-solid products lies in the fact that the API, as opposed to solid dosage forms, is typically present in a dissolved state and can, therefore, undergo chemical reactions more easily. APIs can undergo basic or acidic hydrolysis, various types of oxidation, isomerization, transesterification or migration into or through different types of packaging material. The same also applies to functional excipients such as antioxidants or preservatives whose stability also needs to be monitored. Furthermore, the compatibility of the API with excipients needs to be investigated. As pure excipients may behave differently from the excipient, which is formulated in a complex matrix, two approaches are possible: The stability is first investigated at the level of the formulation, and if an indication for incompatibility with excipients exists, a compatibility screening with potentially responsible excipients is performed. Alternatively, a comprehensive compatibility screening is performed prior to the start of the formulation development. This, however, is associated with the risk of discarding excipient options, which are only problematic when used as pure excipients but not as part of a complex formulation.

Method development should start with stress tests, which are also referred to as forced degradation studies, to obtain information on the stability and degradation behavior of the API when exposed to different stress conditions. Stress tests are mandatory as part of the stability studies for the registration application of new drug substances or drug products as described in the ICH Q 1 A (R2) “Stability Testing of new Drug Substances and Products” [72]. The guideline states that “Stress testing of the drug substance can help identify the likely degradation products, which can in turn help establish the degradation pathways and the intrinsic stability of the molecule and validate the stability-indicating power of the analytical procedures used. The nature of the stress testing will depend on the individual drug substance and the type of drug product involved.” Thermal, oxidative, photolytic and hydrolytic (acidic and basic) stress conditions are required. It is important to be aware that a decrease in the API peak in the chromatographic analysis may not necessarily be connected to a corresponding increase in peaks of degradation products. The earlier a mass balance of the API and its degradation products can be established, the better. For registration, this is mandatory. It is even possible that despite substantial API degradation, there are no peaks of degradation products visible in the chromatograms in case their chromophore is destroyed. Therefore, it is recommendable that detection at multiple wavelengths and/or mass-spectrometric detection is used for stress test studies [69]. The exact experimental conditions are not defined, but potential approaches have been described in the literature [70].

Although this ICH guideline is not binding for early explorative stages of development, it provides useful general guidance. The value generated from properly designed stress tests is in the identification of degradation mechanisms such as hydrolysis, oxidation, thermolysis or photolysis and the establishment of preliminary hypotheses for degradation pathways and possible structures of API-related degradation products (especially if in addition to UV detection, molecular masses are also recorded, e.g., by routinely using an additional single-MS detector). This provides an important basis and good understanding of the API molecule, the identification of options to optimize stability in formulations, and for the development of stability-indicating analytical methods and, ultimately, robust drug products. It may be important to adapt the stress conditions to the API, e.g., to avoid ‘over stress’ resulting in the dominance of secondary degradation products. In such a case, important information would be lost.

Although not a formal requirement in exploratory development, from a risk mitigation point of view, it is important to know if an analytical method is stability-indicating or not as soon as significant investment decisions are made, such as entering the clinical phase or during the process of due diligence for a product to be licensed. This is especially the case if conclusions should be made based on the assay. Without that knowledge, there is a substantial risk of making false-positive decisions on drug product stability. According to Blessy et al. [70], a stability-indicating method (SIM) is an analytical procedure used to quantitate the decrease in the amount of the active pharmaceutical ingredient (API) in drug products due to degradation. A stability-indicating method accurately measures the changes in active ingredient concentration without interference from other degradation products, impurities and excipients. Often, this is confirmed by investigations on spectral homogeneity of the relevant peak(s) in the chromatogram. However, it should be considered that in topical formulations, often complex excipients are used, which may elute over almost the entire range of retention time, as, for example, demonstrated for polysorbate 20 [58]. Alternatively, or in combination with stability-indicating assay methods, a comprehensive analysis of the degradation products based on a good understanding of the degradation kinetics is very helpful in the risk assessment of prototype formulations.

Chemical stability studies in exploratory development are typically performed in the form of accelerated stability studies. As opposed to ICH stability studies, the focus is not on providing data for registration purposes but on predictive risk-analysis and formulation selection under accelerated conditions, i.e., with substantial time savings as compared to real-time studies. Typically, formulations are stored at various temperatures between 5 °C and 40 °C. If the formulations are stable at these temperatures, also 50 °C or even 60 °C may be considered. However, formulations that show major physical changes, such as phase separation or precipitation, should not be included in the analysis as the chemical degradation behavior may be influenced by these physical changes. In exploratory accelerated studies of semi-solid formulations, typically, hermetic packaging materials (e.g., glass vials with screw caps) are used so that controlled humidity is not relevant for the study.

If the reaction constants generated from the stability studies follow the Arrhenius equation, data from 12 week accelerated stability studies often allow a preliminary shelf-life prediction. If a deviation from the Arrhenius equation is observed, a root cause analysis needs to be performed to evaluate if, for example, physical changes in the formulation or a deviation from the assumed reaction kinetic occurred (e.g., incorrect reaction order or subsequent reaction).

If an API is chemically very stable in a formulation and no relevant degradation occurs even at higher temperatures, it will also not be possible to establish Arrhenius calculations. In such a case, however, it will be possible to still assume that achieving a commercial shelf life will be feasible.

The approach to the development of analytical methods should follow a structured and strategic approach already in the early stages of development. Although not a formal requirement in exploratory development, the concept of defining an analytical target profile (ATP) is recommended. According to USP-NF 〈1220〉 “Analytical Procedure life Cycle” [73], an “ATP is a prospective description of the desired performance of an analytical procedure that is used to measure a quality attribute, and it defines the required quality of the reportable value produced by the procedure, aligned with the quality target product profile (QTPP).” The benefits of establishing a preliminary ATP already in the early stages of development are that an explicit reflection on the design goals for a new analytical procedure takes place. The efforts of establishing certain performance attributes of analytical methods can be weighed against the risks of making incorrect decisions when skipping such efforts. Furthermore, an ATP is a useful tool for alignment between product development and other stakeholders or sponsors. In USP-NF 〈1220〉, the ATP is part of the analytical procedure life cycle, which is defined in three stages. Many of the elements of stage 1 are already the best investigated in exploratory development, such as “understanding gained through knowledge gathering, systematic procedure development experiments, and risk assessments and associated lab experiments.” It may be difficult, risky, and require more resources if these elements are investigated only retrospectively. An advantage of the ATP is that the performance criteria of an analytical method are defined rather than a concrete analytical procedure. As a consequence, changes in the concrete procedure of such a method during its life cycle can be approved by the authorities much more easily.

The strategies for stability testing and optimization are highly API and formulation dependent. Apart from the guiding principles outlined above, it is essential not just to follow a general protocol but to work with the generated data and to anticipate potential issues based on the properties of API, formulation, excipients and primary packaging. Compatibility with single excipients may not be predictive of the compatibility at the level of the formulation as dilution or distribution between phases or into the oil-water interface may influence stability. Furthermore, APIs may not react directly with excipients but only with degradation products of excipients or with impurities of excipients, which may be present in higher or lower concentrations, dependent on the batch or age and storage conditions of the excipients. Furthermore, depending on both the properties of the API and the formulation, APIs may migrate into or through the primary packaging or undergo degradation processes catalyzed by certain components of the packaging material.

### 4.3. In Vitro Performance Testing

The classical method for investigating formulation performance is the Franz diffusion cells method established in 1975 [74]. Originally, they were performed as in vitro permeation studies by using skin as a membrane, an experiment established as in vitro permeation testing (IVPT). Alternatively, also filter membranes that only provide a separation of the formulation from the acceptor compartment without providing a relevant barrier for diffusion can be used in an experiment called in vitro release testing (IVRT). Except for the similarities in testing instrumentation, the two methods are not practically comparable as the differences in goals, purposes, and techniques far outweigh the similarities, as summarized in Table 3. Whenever the release of the API from the matrix is a rate-limiting or critical formulation attribute, IVRT may be a suitable tool for performance characterization. In most cases, however, it is rather a quality control tool. The primary readout parameter for IVPT studies is the flux, which is an important parameter for the initial assessment of systemic exposure, which is relevant for safety assessment or transdermal drug delivery. It may also be used as a surrogate for penetration into the skin.

The skin membrane used in the IVPT experiment can also be extracted, and the concentration in different skin layers can be determined. To separate the skin layers, either slicing by a microtome [75] or heat separation of the epidermis and dermis can be used [76]. IVPT and IVRT are described in regulatory guidelines and are important tools for the bioequivalence assessment of topical generic drugs. Comprehensive studies of their use in this context have been published [77,78]. In early development, especially for compound and formulation selection, alternative models can be used. The “Hamburg model of skin penetration” [79] allows the determination of cutaneous biodistribution in viable pig ear skin. For a diverse set of compounds and formulations, it has shown excellent correlation to viable human skin. As the skin remains metabolically viable throughout the experiment, it also offers the advantage that the metabolism of APIs in the skin can be monitored. Apart from the fact that the viability of the skin was not considered, a similar model has been described by Quartier et al. [80]. Dermal open-flow microperfusion and dermal microdialysis, which are more frequently performed as in vivo investigations, have also been demonstrated as potential tools for ex vivo studies on the pharmacokinetics of topically applied drugs [81]. Although the site of action for most topically applied drugs is in the epidermis or dermis, also stratum corneum sampling by tape stripping may be used as a surrogate model [82,83,84]. In recent years, also spectroscopic techniques, mainly by confocal Raman microscopy, have been used for in vitro skin penetration assessment with promising results [9,85,86,87]. This technique can also be used in vivo, which, however, should not be discussed in the context of this work. It should be noted that Raman microscopy is not a universally suitable technique for skin penetration assessment due to limited sensitivity, which means that it may not be able to detect very low concentrations of drugs, e.g., for poorly penetrating compounds or low-dose formulations. Furthermore, the drug molecule needs to have a certain intensity of Raman scattering signal that is distinct from the signals of the skin matrix.

Overall, studying drug release, penetration, or permeation by in vitro models is an important part of the early development of topical semi-solid dosage forms. It is important to understand the potential and limitations of the respective models with regard to the purpose of the investigations. Aspects like the site of action within the skin, detectability, potential metabolism of the API, infinite or finite dosing, and transformation of the vehicle need to be considered.

## 5. Conclusions

During the past decades, progress has been made in the understanding of the design and characterization of semi-solid dosage forms. Furthermore, knowledge of driving forces for skin penetration and the influence of the formulation vehicle in this process has grown. This was also supported by advances in analytical technologies like rheology, spectroscopy as well as chromatography and mass spectrometry for chemical stability studies. These are enablers for a rational design of semi-solid formulations. However, recently launched novel semi-solid medicines, at least in part, appear to follow rather conservative and potentially empirical formulation approaches. A reason for that may be the strong impact of aspects of product quality in the strategy and risk management of topical development projects in combination with increased risks of physical and chemical instability as compared to peroral dosage forms.

An approach to rational formulation design based on a systematic formulation design around the specific physicochemical properties of an API can help to combine the requirements for quality with innovative formulation approaches and optimization of aesthetic appeal and performance of formulations. The better the underlying design principles of a formulation—such as the solubility, saturation, and distribution of the API and the critical excipients—are understood, the better its critical quality attributes can be assessed, and the more targeted the formulation can be optimized and modified if necessary.

## Figures and Tables

**Figure 1 pharmaceutics-15-01822-f001:**
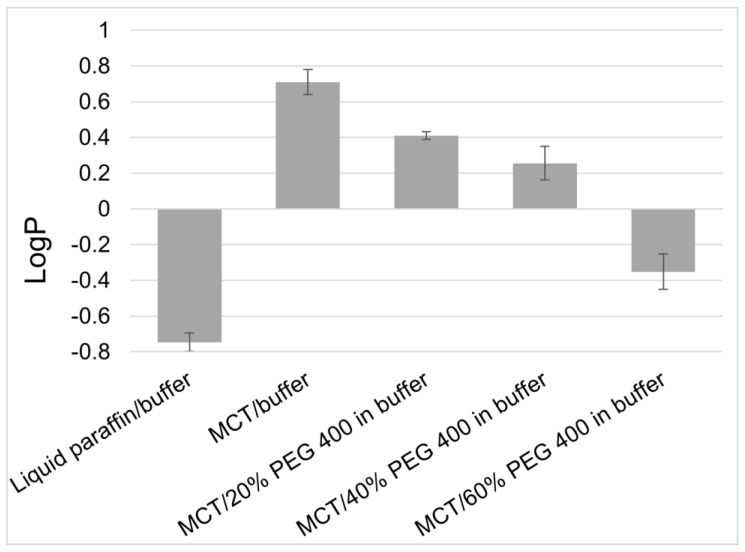
Logarithmic distribution coefficients (LogP) of phenoxyethanol between different oil and water phases. Adapted from reference [50].

**Figure 2 pharmaceutics-15-01822-f002:**
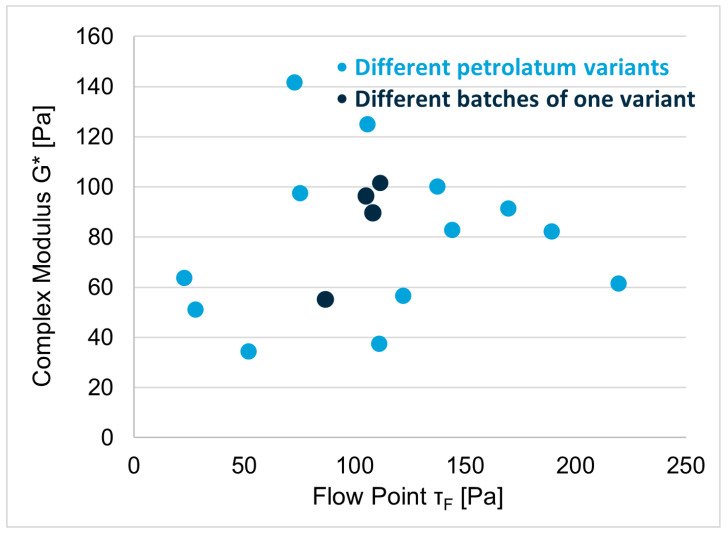
Complex moduli and flow points of 13 different petrolatum variants (**blue dots**) and of four different batches of the same variant (**black dots**). All variants we compliant with Pharmacopeia Eur. specifications. Adapted from reference [3].

**Figure 3 pharmaceutics-15-01822-f003:**
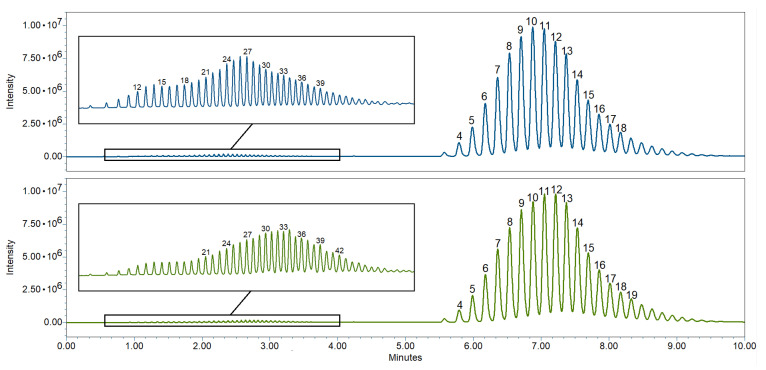
Chromatograms of lauromacrogol 400 from two different suppliers. Differences exist in the number of ethylene oxide moles per mole lauryl alcohol of the polyoxyethylene ethers (**right part** of the chromatogram) and in the composition of free polyoxyethylene chains (insert on the **left**). The numbers above the chromatographic peaks indicate the number of polyoxyethylene units.

**Figure 4 pharmaceutics-15-01822-f004:**
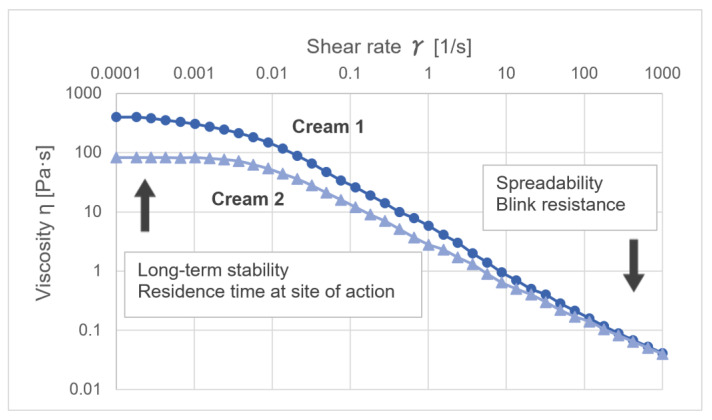
Differences in shear-thinning behavior of two creams. While both creams show comparable viscosity at higher shear rates, cream 1 (dots) has a 5-times higher zero-shear viscosity than cream 2 (triangles).

**Figure 5 pharmaceutics-15-01822-f005:**
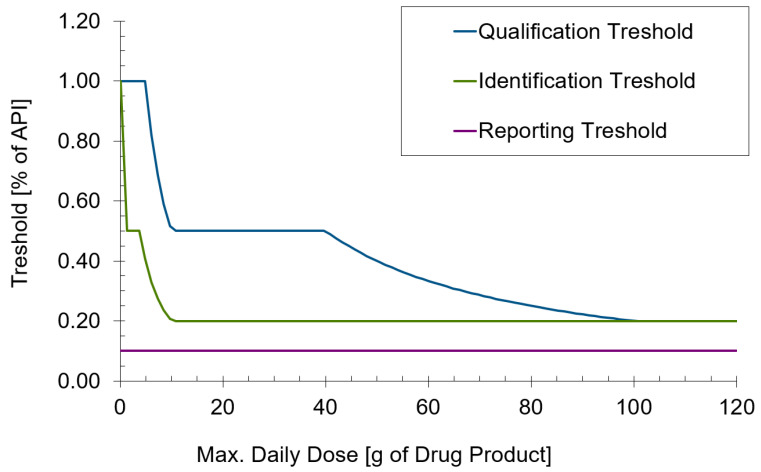
Illustration of the dependency of qualification, identification, and reporting thresholds for impurities dependent on the maximum daily dose (MMD) according to ICH Q3B [71].

**Table 1 pharmaceutics-15-01822-t001:** Overview of qualitative formulation composition of semi-solid topical medicines based on NCE products approved by the FDA from 2016–2022 [22]. The “x” indicates that the respective excipient is contained in the formulation.

Trade Name		Eucrisa^®^	Aklief^®^	Winlevi^®^	Klisyri^®^	Opzelura^®^	VTAMA^®^	ZORYVE^®^
API and Concentration		Crisaborole 2%	Trifarotene 0.005%	Clascoterone 1%	Tirbanibulin 1%	Ruxolitinib 1.5%	Tapinarof 1%	Roflumilast 0.3%
Indication		Atopic Dermatitis	Acne Vulgaris	Acne Vulgaris	Actinic Keratosis	Atopic Dermatitis	Plaque Psoriasis	Plaque Psoriasis
Formulation Type		Ointment	Cream	Cream	Ointment	Cream	Cream	Cream
Year of US Approval		2016	2019	2020	2020	2021	2022	2022
Excipient	Properties/Typical Function							
White petrolatum	Ointment base/oil phase	x				x		x
Paraffin	Oil phase	x						
Light mineral oil	Oil phase					x		
Mineral oil	Oil phase			x				
Medium-chain triglycerides	Oil phase		x			x	x	
Isopropyl palmitate	Oil phase							x
Cyclomethicone	Oil phase		x					
Dimethicone 350	Oil phase					x		
Polyethylene glycol 200	Solvent					x		
Hexylene glycol	Solvent							x
Propylene glycol	Solvent/penetration enhancer	x	x	x	x	x	x	
Diethylene glycol monoethyl ether	Solvent/penetration enhancer						x	x
Purified water	Hydrophilic solvent		x	x		x	x	x
Ethanol	Hydrophilic solvent		x					
Emulsifying wax	Emulsifier/consistency agent						x	
Glyceryl stearate SE	Emulsifier/consistency agent					x		
Mono- and di-glycerides	Consistency agent/co-emulsifier	x		x	x			
Cetyl alcohol	Consistency agent/co-emulsifier			x		x		
Cetostearyl alcohol	Consistency agent/co-emulsifier							x
Stearyl alcohol	Consistency agent/co-emulsifier					x		
Polysorbate 20	Emulsifier/solubilizer					x		
Polysorbate 80	Emulsifier/solubilizer			x			x	
Polyoxyl 2 stearyl ether	Emulsifier						x	
Polyoxyl 20 stearyl ether	Emulsifier						x	
Ceteareth-10 phosphate	Emulsifier							x
Cetearyl phosphate	Emulsifier							x
Xanthan gum	Thickener					x		
Copolymer acrylamide and acryloyldimethyl taurate *	Thickening and emulsifying polymer		x					
Allantoin	Humectant		x					
Phenoxyethanol	Preservative		x			x		
Methylparaben	Preservative					x		x
Propylparaben	Preservative					x		x
Benzoic acid	Preservative						x	
Butylated hydroxytoluene	Antioxidant	x					x	
Vitamine E	Antioxidant			x				
Edetate calcium	Chelating agent	x						
Edetate disodium	Chelating agent				x	x		
Citric acid monohydrate	Buffer			x			x	
Sodium citrate dihydrate	Buffer						x	
Sodium hydroxide	pH adjustment							x

* dispersion 40% in isohexadecane.

**Table 2 pharmaceutics-15-01822-t002:** Selected solubility data of tacrolimus in different solvents and solvent mixtures. Adapted from reference [32].

Solvent	Solubility of Tacrolimus [mg/mL]
**Water and aqueous solutions**	
Water	<0.001
**Co-Solvents**	
Triethyl citrate	105.2
**Oil phases**	
Medium chain triglycerides (MCT)	10.8
Diisopropyl adipate (DIPA)	63.3
Isopropyl myristate	4.1
Liquid paraffin	0.007
**Composed oil phases**	
DIPA:MCT (50:50)	26.4
DIPA:paraffin:MCT (25:25:50)	10.1

**Table 3 pharmaceutics-15-01822-t003:** Differences between in vitro release testing (IVRT) and in vitro permeation testing (IVPT).

Parameter	IVRT	IVPT
Investigated process	Release	Permeation
Membrane used	Synthetic filter membrane	(human) skin
Donor chamber exposure	Occluded	Often unoccluded
Dosing	Infinite dose	Often (semi)finite dose
Readout parameters	Flux profile (Jmax, etc.)	Release rate (slope)
Receptor cell media	Non-physiological media acceptable	Physiological media preferred
Typcial detection range	µg/mL range	ng/mL range
Contact of product with acceptor	Product-media interface	Product stays “dry“
Variability in membrane	Relatively consistent in quality	Donor variability
in vitro/in vivo correlation	Not designed to correlate with in vivo	IVIV correlation expected
Overall purpose	Assessment of quality	Assessment of performance/safety

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
