# Peer review of "Rational Design of Topical Semi-Solid Dosage Forms-How Far Are We?"

_pharmaceutics, 2023, doi:10.3390/pharmaceutics15071822_

Round 1

Reviewer 1 Report

1.    Few references must be cited in the first and second paragraph of the introduction part.

2.    The reference for “International Council for Harmonisation of Technical Requirements for Pharmaceuticals for Human Use” guideline must be cited.

3.    Typically, a shelf life of at least two years at room temperature is required for reasons of commercial viability for topical drug products (the exact conditions for stability testing depend on the respective climate zone as per WHO definitions), Reference?

4.    One attribute that is always specified is the assay of the active ingredient(s) (API). The standard limit is 95%-105% of the declared concentration in Europe and 90%-110% in the US, Reference?

5.    Specifications for physical stability are derived from critical quality attributes (CQA) and may include pH, viscosity, and particle or globule size distribution for semisolid product, Reference?

6.    As the active tirbanibulin is contained at 1%, it can be concluded that mono- and di-glycerides are present at 10%, so that the full quantitative composition is available in this case, write down the %, w/w or w/v.

7.    The title “Rational Design of Topical Semisolid Dosage Forms - How Far Are We?” itself indicating the formulations are meant for topical application, the authors must give some emphasis on the route of administrations (topical/ transdermal/ ocular etc.) also.

8.    Although it is a good literature review, the authors must put their own opinion at each and every section to make it good “Review Article”.

9.    “Overall, studying drug release, penetration, or permeation by in vitro models is an important part of early development of topical semisolid medicines”, it should be semisolid dosage forms not medicines. Also, why not the in vivo animal model is also an important parameter of early development of topical semisolid dosage forms?

10. It would be far better if the information given in the section 4.2 Chemical analytics of semisolids in a Tabular form for better and easy understanding to the readers.

11. The muco-or bio-adhesion is an important parameter during the characterization of semisolid formulations, which was not discussed in this manuscript.

12. Also, if the product is meant for ophthalmic application, the transparency is also an important parameter, it should be also considered.

Minor editing of English language required. 

Author Response

Thank you for your comments. Please see the attachment for our responses.

Reviewer 2 Report

The manuscript written by Herbig et al. is about discussing the role of semi-solid formulations in topical administrations. This is an exceptionally good review, which is rather formative and interesting. The topic highly fits within the journal’s scope, and I recommend the publication of this manuscript in Pharmaceutics though the authors should consider the following suggestions/revisions:

Minor comments

(1) The authors should include the role of the “quality by design (QbD)” approach for developing pharmaceutical formulations

(2) The authors should provide more critical analysis in their conclusions

The Quality of English language is fine

Author Response

(The authors gave the same response as above.)

Reviewer 3 Report

The authors provide a comprehensive description of the current status and challenges of semi-solid formulation research for topical application, which was considered to bring useful information in a general overview.

Author Response

Thank you for your comments. We understand that no specific amendments of our manuscript were requested.

Round 2

Reviewer 1 Report

The authors have revised the manuscript as per the comments and suggestion.